# Killer prey: Ecology reverses bacterial predation

**Marie Vasse** [1]*, **Francesca Fiegna**[2], **Ben Kriesel**[2], **Gregory J. Velicer**[2]*

**1** MIVEGEC (UMR 5290 CNRS, IRD, UM), CNRS 34394 Montpellier, France, **2** Institute for Integrative Biology, ETH Zürich, Zürich, Switzerland

* contact@marievasse.eu (MV); gregory.velicer@env.ethz.ch (GJV)

**Data Availability Statement:** The R script used to analyze this data and the datasets are available on Github at https://github.com/marievasse/Killer-prey and Zenodo (10.5281/zenodo.10214013).

## Abstract

Ecological variation influences the character of many biotic interactions, but examples of predator–prey reversal mediated by abiotic context are few. We show that the temperature at which prey grow before interacting with a bacterial predator can determine the very direction of predation, reversing predator and prey identities. While *Pseudomonas fluorescens* reared at 32°C was extensively killed by the generalist predator *Myxococcus xanthus*, *P. fluorescens* reared at 22°C became the predator, slaughtering *M. xanthus* to extinction and growing on its remains. Beyond *M. xanthus*, diffusible molecules in *P. fluorescens* supernatant also killed 2 other phylogenetically distant species among several examined. Our results suggest that the sign of lethal microbial antagonisms may often change across abiotic gradients in natural microbial communities, with important ecological and evolutionary implications. They also suggest that a larger proportion of microbial warfare results in predation—the killing and consumption of organisms—than is generally recognized.

## Introduction

Ecological context strongly shapes the intensity and character of many biotic interactions [1,2], including between predators and prey [3]. Some ecological factors influence predator–prey interactions contemporaneously. For example, water temperature immediately influences the timing and duration of predator attacks in some fish [4], background versus prey coloration often determines prey detectability [5], and snow depth modulates the efficiency of wolf predation [6]. Most such examples illustrate how ecological context modulates the effectiveness of predation in a unidirectional predator–prey relationship. However, at least 1 study has found that abiotic ecology can even reverse the predominant direction of mutual predation; specifically, the majority direction of predation events between 2 amphipod species has been shown to reverse as a function of salinity [7].

Ecological effects on predator–prey interactions can also be delayed. For example, maternal exposure to predation risk at one time can influence the degree of antipredator behavior displayed by offspring at a later time [8]. Investigating how predator–prey interactions change as a function of ecological context, whether contemporaneous or historical, is necessary for understanding how those interactions evolve in spatially and temporally heterogeneous habitats.

**Funding:** This work was funded by Swiss National Science Foundation (SNSF) grant 310030B_182830 to GJV. The funders had no role in study design, data collection and analysis, decision to publish, or preparation of the manuscript.

**Competing interests:** The authors have declared that no competing interests exist.

Predation pervades the microbial world [9] as well as animal communities, impacting microbial community composition and network structure and thereby influencing nutrient-cycling dynamics and important features of macroorganism biology. Behavioral modes of microbial predation vary greatly. Many eukaryotic predators of bacteria such as nematodes and many protists ingest whole prey cells [10,11], while predatory bacteria are too small to do so. In a virus-like life cycle, cells of the specialist bacterial predator *Bdellovibrio bacteriovorus* physically attach to a prey cell before invading it and reproducing inside [12]. Other bacteria kill prey without attachment using diffusible secretions [13,14]. In addition to using diffusible killing agents, the generalist bacterial predator *Myxococcus xanthus* employs highly effective predatory weapons that depend on contact with prey cells [15,16]. Perhaps best known for its formation of fruiting bodies in response to starvation [17], *M. xanthus* can prey on a very broad range of other microbes, including both gram-negative and gram-positive bacteria [18] and some fungi [19]. *M. xanthus* uses 2 synergistic motility systems to forage for prey across variable solid surfaces [20].

Heterogeneity in many abiotic factors such as temperature, prey nutrition, pH, oxygen availability, salinity, and surface or fluid viscosity is known or likely to influence microbial predator–prey interactions [21–26], and microbial antagonisms more broadly [27–30]. For example, predators and prey may be differentially sensitive to pH gradients [25,31] and temperature influences many bacterial traits relevant to predation, including cell division rates and motility behavior [32,33]. Temperature also impacts secondary metabolite production [34,35] and Type VI secretion [36,37], traits associated with a broad range of microbial antagonisms often not described as predatory, and can modulate the intensity of such antagonisms [38].

Motivated by observations suggesting that *M. xanthus* predation could differ greatly on prey reared on a room temperature bench versus in a warmed incubator, we investigated how variation in the temperature at which bacterial prey grow prior to predator attack might alter the predation risk of prey. Using a phylogenetically diverse panel of bacteria previously demonstrated to fuel predatory growth by *M. xanthus* [18,39], we initially quantified the ability of *M. xanthus* to swarm through prey lawns reared at 3 temperatures (12, 22, or 32˚C). Upon observing that *Pseudomonas fluorescens* eliminated *M. xanthus* swarming when the former had been previously reared at 22˚C but not at 32˚C, we quantified *M. xanthus* viable population size after interaction with *P. fluorescens* pregrown at the same 3 temperatures from several initial densities. Interaction with *P. fluorescens* grown at 22˚C resulted in *M. xanthus* extinction. We then tested whether the killing factors produced by this sometime-prey species are cell-bound or diffusible secretions and whether they are proteinaceous or not (or require polypeptides to function) by testing for functional sensitivity to 95˚C heat exposure. We further asked whether *P. fluorescens* functions as a predator after killing *M. xanthus* by testing whether nutrients from killed *M. xanthus* fuel *P. fluorescens* growth. Finally, we tested whether *P. fluorescens* secretions lethal to *M. xanthus* also kill a diverse panel of other potential prey species.

## Materials and methods

### Strains

*M. xanthus* strains GJV1, DK3470, A75, and Serengeti 01 were used in this study. GJV1 is a closely related derivative of the reference strain DK1622 [40], DK3470 is a mutant of DK1622 with a mutation in the *dsp* gene [41], and A75 [42] and Serengeti 01 [43] are natural isolates from Tübingen, Germany, and Serengeti National Park, Tanzania, respectively. Strains of *Arthrobacter globiformis*, *Bacillus bataviensis*, *Curtobacterium citreum*, *Escherichia coli*,

*Micrococcus luteus*, and *Rhizobium vitis* used here are the same as those reported by Morgan and colleagues in 2010 [18]. *Pseudomonas fluorescens* strain SBW25 is from [44].

## General media and culture conditions

Unless specified otherwise, *M. xanthus* strains were inoculated from frozen stock onto CTT 1.5% agar (1% casitone; 8 mM $MgSO_4$; 10 mM Tris–HCl (pH 7.6); 1 mM $KH_2PO_4$, 1.5% agar [45]) plates 3 days prior to transfer of culture samples into 8 ml CTT liquid (identical to CTT agar except lacking agar) in 50-ml flasks. Liquid cultures were typically grown over 2 days with a dilution transfer after 1 day. All assays were initiated from log-phase cultures. For other species, samples from frozen stocks were inoculated directly into LB liquid. Unless specified otherwise, liquid cultures were incubated at 32˚C with shaking at 300 rpm and plate cultures were incubated at 32˚C, 90% relative humidity. Cell densities of bacterial populations were estimated with a TECAN Genios plate reader. Prior to resuspension to initiate assays, cultures were centrifuged at $4,472 \times g$, 15 minutes. Unless specified otherwise, all bacterial cultures were resuspended in M9 medium (1×M9 salts: 22 mM $KH_2PO_4$, 18.7 mM $NH_4Cl$, 8.6 mM NaCl supplemented with 2 mM $MgSO_4$ and 0.1 mM $CaCl_2$), and experiments were run on M9cas agar (M9 medium supplemented with 0.3% casitone and 1.2% agar).

## Swarming rates

Around 25-ml aliquots of M9cas agar were poured into 9-cm petri plates and allowed to cool and solidify in a laminar flow hood without lids for 20 minutes, after which they were capped and stored overnight at room temperature.

Centrifuged prey cultures were resuspended to a predicted density of approximately $5 \times 10^6$ cells/ml in M9 medium. From each resuspended culture (and one control containing only M9), 600-μl aliquots were placed on M9cas agar plates and distributed evenly with a sterile metal triangle. Plates were then left open without lids in a laminar flow hood for 60 minutes. Four plates for each prey species were then incubated at each of 3 temperatures (12, 22, and 32˚C) for 22 hours. Incubator windows were covered to prevent light penetration. After incubation, the plates were kept at room temperature for 2 hours prior to addition of *M. xanthus*.

Centrifuged cultures of *M. xanthus* strains GJV1, A75 and Serengeti 01 were resuspended to a predicted density of approximately $10^{10}$ cells/ml with M9 medium. For each temperature–prey combination (and for control plates), 20 μl of the resuspension were spotted in the middle of the plate and plates were then left open without lids in a laminar flow hood for 30 minutes before being incubated at 32˚C and 90% relative humidity. *M. xanthus* swarm diameters were measured after 7 days of incubation (2 perpendicular diameters per swarm at random orientation). *M. xanthus* swarms of these strains are bright yellow in color. When no evidence of *M. xanthus* growth was observed (i.e., no yellow area) after 7 days even within the originally inoculated plate area, a diameter value of 0 was recorded. (This occurred only on some plates with *P. fluorescens*.)

## Test for *M. xanthus* killing of *P. fluorescens* reared at 32˚C

For our test of whether *M. xanthus* kills *P. fluorescens* reared at 32˚C on M9cas agar, media and culture-handling protocols were the same as in the swarming assays described above except in the following respects. The killing test was performed on M9cas agar in 50-ml glass flasks rather than in petri dishes to allow shaking with resuspension buffer (see details below). A 10-μl aliquot of resuspended *P. fluorescens* culture was inoculated onto M9cas agar the day before addition of *M. xanthus*. The aliquot was not spread across the plate (as the 600-μl aliquots in the swarming assays were) but was rather allowed to grow into a small circular lawn

within the originally inoculated spot area. After incubation for 24 hours at 32˚C, a 50-µl aliquot of *M. xanthus* resuspension (approximately $10^{10}$ cells/ml, as in the swarming assays) was inoculated across the top of and immediately surrounding the circular *P. fluorescens* lawn that had grown up overnight. For the control treatment to which *M. xanthus* was not added, a 50-µl aliquot of M9 liquid was added in the same manner. The flasks were then incubated for 4 days before *P. fluorescens* was harvested and dilution plated into LB 0.5% soft agar, in which *M. xanthus* does not grow. *P. fluorescens* was harvested by adding 1-ml M9 liquid to each flask, scraping the bacteria into suspension with a loop and mixing the suspension by repeated pipetting to disperse *P. fluorescens* cells. To suspend any cells remaining on the agar after the above procedure, 2 ml of additional M9 liquid and 10 glass beads were then added to each flask and flasks were shaken for 1 hour at 300 rpm, 32˚C prior to dilution plating.

## Test for effects of *P. fluorescens* rearing temperature and inoculum population size on *M. xanthus* DK3470 survival

One day prior to inoculation, 10-ml aliquots of M9cas agar were poured into 50-ml glass flasks and allowed to solidify without flask covers for 30 minutes in a laminar flow hood. Flasks were then capped and kept at room temperature overnight. Centrifuged cultures of *P. fluorescens* were resuspended to predicted densities of approximately $5 \times 10^5$, approximately $5 \times 10^6$, and approximately $5 \times 10^7$ cells/ml with M9 liquid. A 200-µl aliquot of resuspended culture was inoculated into each agar flask and then spread across the agar surface by gentle rotation. Flasks were then kept open without covers for 60 minutes in a laminar flow hood, after which they were capped and flasks of each inoculum population size treatment were incubated at 3 temperatures (12, 22, and 32˚C) for 22 hours. After incubation at different temperatures, all flasks were kept at room temperature for 2 hours prior to either addition of *M. xanthus* or assessment of *P. fluorescens* population size.

For assays of *M. xanthus* strain DK3470 population size, 20-µl aliquots of DK3470 culture resuspended to approximately $10^{10}$ cells/ml (approximately $2 \times 10^8$ cells) were spotted in the middle of each flask; flasks were then left open for 30 minutes in a laminar flow hood. Flasks were then harvested at one of 2 time points: either immediately or after 7 days of incubation at 32˚C. To harvest and disperse DK3470, 10 glass beads and 1-ml M9 liquid were added to each flask and flasks were shaken at 300 rpm, 32˚C for 15 minutes before the resulting suspensions were dilution plated into CTT 0.5% soft agar containing gentamicin (10 µg/ml), which prevents growth of *P. fluorescens* but not *M. xanthus*. Colonies were counted after 7 days of incubation at 32˚C, 90% relative humidity.

## Test for effect of *P. fluorescens* inoculum population size on postgrowth population size

For assays of *P. fluorescens* population size after overnight growth at one of 3 temperatures, assays that were performed separately from the above-described assays with DK3470, 10 glass beads and 1 ml of M9 liquid were added and the flasks shaken at 300 rpm, 32˚C for 15 minutes to disperse *P. fluorescens* populations. Samples were then dilution plated into LB 0.5% soft agar, and *P. fluorescens* colonies were counted after 2 days of incubation at 32˚C, 90% relative humidity.

## Test for diffusion of the killing compound(s) produced by *P. fluorescens*

DK3470 was inoculated onto CTT hard agar from frozen stock 4 days prior to inoculation in 8-ml CTT liquid in a 50-ml Erlenmeyer flask. The resulting liquid culture was grown for

approximately 24 hours at 32°C, 300 rpm, diluted into fresh medium and grown for another 24 hours before being centrifuged and resuspended to approximately $5 \times 10^9$ cells/ml in either supernatant from buffer suspensions of *P. fluorescens* (prepared as described below) or control buffer. The resuspended cultures were incubated for 6 hours at 32°C and then dilution plated into CTT 0.5% soft agar. Plates were incubated for 3 to 5 days before colonies were counted.

To prepare *P. fluorescens* supernatants, *P. fluorescens* was inoculated from frozen stock into 8-ml LB liquid and then incubated overnight at 32°C, 300 rpm. After dilution to approximately $5 \times 10^7$ cells/ml, 200-μl aliquots of the resulting culture were spread across the surfaces of 10-ml aliquots of M9cas agar that had been poured into 50-ml flasks 24 hours before. *P. fluorescens* populations and control flasks containing only M9cas agar were then incubated at 12, 22, and 32°C. After approximately 24 hours, 0.8-ml M9 liquid and approximately 10 sterile glass beads were added to flasks, which were then shaken at 300 rpm, 32°C for 15 minutes before 800 μl of each of the resulting culture suspensions were removed from the flask and centrifuged (5,000 rpm, 15 minutes). After centrifugation, supernatant was filter sterilized with 0.2-μm filters. Each sample of filtered supernatant was separated into 2 equal subsamples, one of which was heated at 95°C for 45 minutes while the other was kept at room temperature.

## Test for effects of *P. fluorescens* on *M. xanthus* when pregrowth of both species and their interaction all occurred at either 22 or 32°C

One day prior to inoculation, 10-ml aliquots of M9cas agar were poured into 50-ml flasks and allowed to solidify for 60 minutes in a laminar flow hood, after which flasks were capped and kept at room temperature overnight. Centrifuged cultures of *P. fluorescens*, previously grown in LB liquid at 22°C or 32°C, were resuspended to predicted densities of approximately $5 \times 10^7$ cells/ml with M9 liquid. A 200-μl aliquot of resuspended culture was spread across the agar surface in each flask by gentle rotation. As a control treatment, 200 μl M9 liquid was spread instead of *P. fluorescens* culture. Flasks were then kept open without covers for 60 minutes in a laminar flow hood, after which they were closed and each was incubated for approximately 28 hours at the same temperature as its source liquid culture had been grown.

DK3470 previously grown on CTT agar was grown in CTT liquid at either 22 or 32°C for approximately 2 days before 20-μl aliquots resuspended to approximately $1 \times 10^{10}$ cells/ml were placed on agar cultures of *P. fluorescens* that had been incubated at the same temperature as the corresponding *M. xanthus* culture. Flasks were then left open for 30 minutes in a laminar flow hood before being incubated for 24 hours at the same temperature as both species in the flask had been incubated prior to interaction. Cultures were then resupended and dilution plated into CTT 0.5% soft agar with gentamicin (10 μg/ml) to count DK3470 population sizes.

## Tests for growth of *P. fluorescens* on nutrients from *M. xanthus* killed by *P. fluorescens* and for effects of *P. fluorescens* rearing temperature on *M. xanthus* DK3470 and *P. fluorescens* population sizes after interaction

One day prior to inoculation, 0.5-ml aliquots of M9cas agar were placed into 48-well plate wells and allowed to solidify for 60 minutes in a laminar flow hood at room temperature, after which the plates were covered and kept at room temperature overnight. Centrifuged cultures of *P. fluorescens* were resuspended to predicted densities of approximately $5 \times 10^5$ cells/ml with M9 liquid. A 10-μl aliquot of resuspended culture was inoculated into each well with agar and then spread across the agar surface by gentle rotation. Plates were then kept open for 60 minutes in a laminar flow hood, after which they were closed and incubated at 22 or 32°C for approximately 24 hours. To estimate *P. fluorescens* population size after the 24 hours of growth but prior to addition of *M. xanthus*, lawns grown at each temperature were harvested by

pipetting-resuspension with 0.5-ml M9 liquid, and then samples were dilution plated into LB 0.5% soft agar.

To initiate interaction between *M. xanthus* and *P. fluorescens* after the 24-hour period of *P. fluorescens* lawn growth, overnight *M. xanthus* DK3470 cultures were centrifuged and resuspended in M9 liquid to a density of approximately $2 \times 10^{10}$ cells/ml. One sample of each resuspension was dilution plated into CTT 0.5% soft agar containing gentamicin (10 μg/ml) to estimate viable population size. Twenty μl aliquots of resuspended *M. xanthus* cultures or of M9 liquid as controls were spotted in each well on top of the *P. fluorescens* lawn; plates were then left open for 30 minutes in a laminar flow hood before being incubated at 32˚C, 90% relative humidity for another 24 hours.

Wells were harvested after 24 hours by adding 0.5 ml of M9 liquid and resuspended by pipetting; the resulting suspensions were dilution plated into LB 0.5% soft agar (which allows growth of only *P. fluorescens*) and CTT 0.5% soft agar containing gentamicin (10 μg/ml, which allows growth of only *M. xanthus*). Colonies were counted after 3 or 7 days at 32˚C, 90% relative humidity for *P. fluorescens* and *M. xanthus*, respectively.

## Test for growth of *P. fluorescens* on nutrients <0.2 μm in size derived from *M. xanthus* killed by *P. fluorescens*

To test if *P. fluorescens* could grow on readily diffusible nutrients released by *M. xanthus* incubated in *P. fluorescens* supernatant, *P. fluorescens* supernatant was harvested following the same protocol described above, except flasks with *P. fluorescens* lawns were incubated overnight only at 22˚C. *M. xanthus* DK3470 cells from exponential phase cultures were spun down and resuspended to a density of approximately $5 \times 10^9$ cells/ml in the *P. fluorescens* supernatant. These resuspensions with *M. xanthus*, as well as supernatant without *M. xanthus*, were then incubated for 6 hours, after which (i) 10-μl samples from the treatment with DK3470 were plated onto CTT 1.5% hard agar to test for *M. xanthus* growth after subsequent incubation and (ii) 200 μl of M9 liquid culture of *P. fluorescens* at densities of approximately $10^6$ and approxiamtely $10^8$ cells/ml were added to both supernatant treatments (with and without DK3470 resuspended cells). The resulting *P. fluorescens* cultures were then dilution plated onto LB 1.5% soft agar immediately and after 24, 48, and 96 hours to determine population size.

## Test for *P. fluorescens* supernatant effects on other bacterial species

Erlenmeyer flasks with M9cas agar were prepared as in previous experiments. Centrifuged cultures of *P. fluorescens* were resuspended to a predicted density of $5 \times 10^7$ cells/ml with M9 liquid, and 200 μl of resuspended samples (or of M9 liquid for the controls) were inoculated into each agar flask and then spread over the agar surface by gentle rotation. Flasks were subsequently kept open for 60 minutes in a laminar flow hood before being closed and incubated at 22˚C for 22 hours. To harvest *P. fluorescens*, 10 glass beads and 0.8 ml of M9 liquid were added to flasks, which were then shaken at 300 rpm, 32˚C for 15 minutes. Suspensions were then centrifuged to harvest supernatants, which were then sterilized with a 0.2-μm filter.

Centrifuged cultures of *A. globiformis*, *B. bataviensis*, *C. citreum*, *E. coli*, *M. luteus*, *M. xanthus* strain DK3470, *P. fluorescens*, and *R. vitis* were resuspended to a predicted density of $5 \times 10^9$ cells/ml with the filtered supernatant, or M9 liquid in the controls. The resulting cell suspensions were incubated at 32˚C, 90% relative humidity for 6 hours before they were dilution plated into CTT (*M. xanthus*) or LB (other species) 0.5% soft agar. Colonies were counted after 2 to 4 days of incubation.

### pH of *P. fluorescens* supernatant

Supernatants from *P. fluorescens* cultures on agar medium incubated at 22°C and 32°C were harvested as previously described. pH values of the supernatants and of M9 medium were measured with a Mettler Toledo pH meter for 3 independent replicate cultures for each temperature treatment.

### Statistical analysis

All analyses were performed using R 4.2.3 [46] as implemented in Rstudio version 2023.03.0 +386 [47]. We used linear models followed by type III ANOVA (package car [48]) and post hoc contrasts (packages emmeans [49] and multcomp [50]) when appropriate. Figures were created using the package ggplot2 [51]. The R script used to analyze this data and the datasets are available on Github at https://github.com/marievasse/Killer-prey and Zenodo (10.5281/ zenodo.10214013).

## Results

### *M. xanthus* swarming on *P. fluorescens* lawns depends on prey-rearing temperature

We first tested whether varying the temperature at which potential prey bacteria grow prior to interacting with *M. xanthus* would impact the rate at which *M. xanthus* populations swarm through prey lawns at a single temperature after predator release. Previously, the rate at which *M. xanthus* swarms through prey lawns on an agar surface has been shown to correlate with several other measures of predatory performance [39]. We performed these assays on M9cas agar containing 0.3% casitone, which *M. xanthus* can use to fuel growth in the absence of prey. However, if prey are first grown in lawns on M9cas before *M. xanthus* is added, the prey sequester nutrients from the casitone into prey-cell biomass and *M. xanthus* cells can potentially fuel growth from prey cells by killing and consuming them. *M. xanthus* might also fuel some growth from any residual casitone nutrients not consumed by prey.

Lawns of 6 diverse bacterial species known to serve as prey to *M. xanthus* (*A. globiformis*, *B. bataviensis*, *E. coli*, *M. luteus*, *P. fluorescens*, and *R. vitis* [39]) were reared on M9cas agar for 22 hours at 12, 22, or 32°C and then all incubated at room temperature for 2 hours prior to release of *M. xanthus*. Aliquots of 3 *M. xanthus* strains (GJV1, A75, and Serengeti 01; 20 μl at approximately $10^{10}$ cells/ml) were spotted at the centers of the prey lawns and allowed to grow and swarm outward to their ability for 7 days at 32°C before swarm diameters were measured.

Across the entire dataset, swarm sizes were impacted by prey presence, prey identity, *M. xanthus* genotype and the rearing temperature of prey, as well as interactions among these factors ($F_{62,171}$ = 38.08, $p$ < 0.01; S1A Table). Swarming rates in the absence of any prey were faster than swarming across prey lawns (Fig 1, Tukey-adjusted contrasts averaged over interaction terms, all $p$ < 0.01).

Prey-rearing temperature had no effect on the swarming rates of any *M. xanthus* genotype for 5 prey species but strongly impacted swarming of all *M. xanthus* genotypes on lawns of *P. fluorescens* (Tukey-adjusted contrasts for each predator–prey combination; S1B Table). All 3 *M. xanthus* genotypes swarmed effectively when *P. fluorescens* had grown at 32°C, swarmed relatively poorly when the prey were grown at 12°C, and did not grow detectably at all when the prey were grown at 22°C. Thus, the temperature at which *P. fluorescens* prey lawns grew prior to encountering a bacterial predator strongly determined the predator's ability to swarm through those lawns.

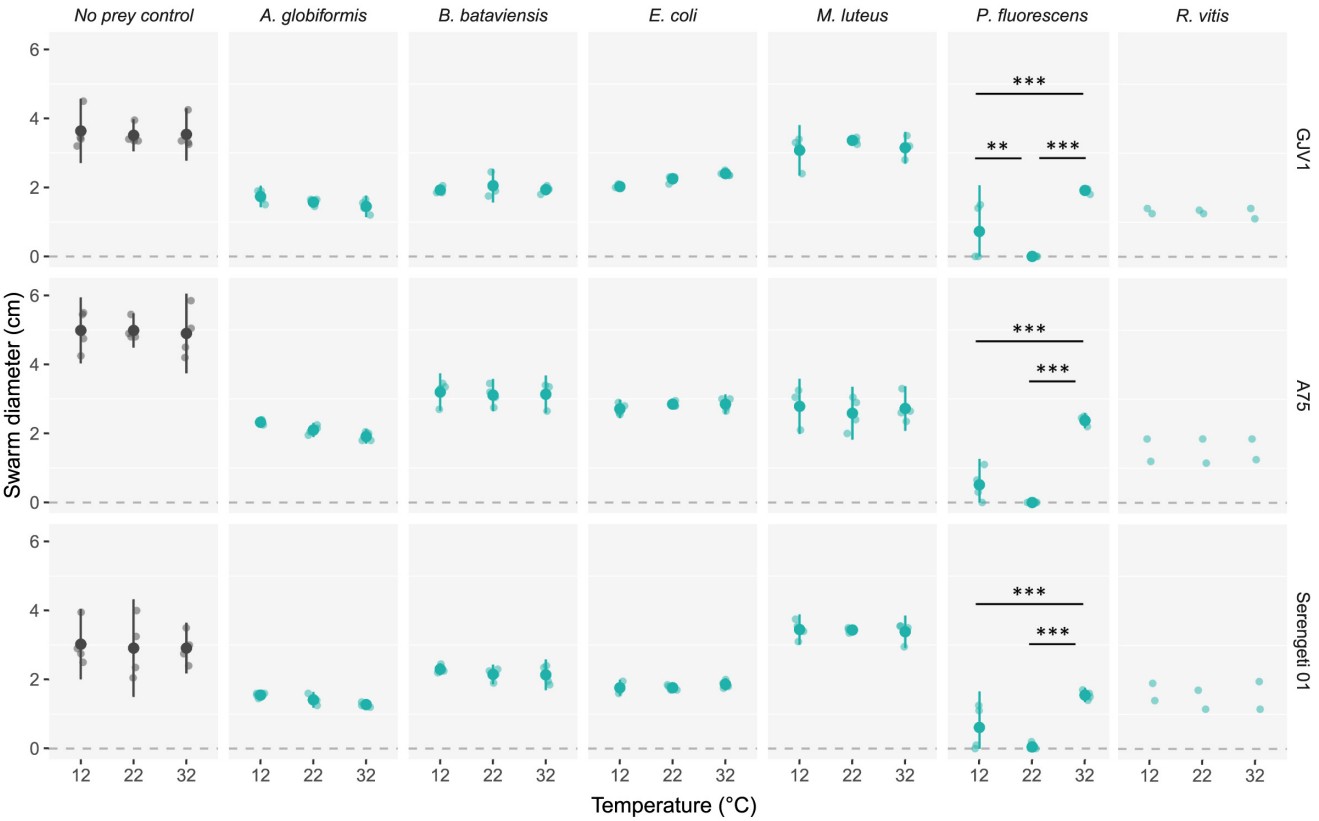

**Fig 1. *M. xanthus* swarming through *P. fluorescens* lawns depends on the temperature of *P. fluorescens* growth prior to predator–prey interaction.** Swarm diameters of 3 *M. xanthus* genotypes (rows) after 7 days on M9cas agar bearing either a lawn of one of several prey species (green dots) or no prey (black dots). Prey lawns were incubated at 12, 22, or 32°C for 22 hours and then brought to room temperature for 2 hours before *M. xanthus* was added. Small dots are biological replicates ($n = 3$ except for *R. vitis* for which $n = 2$), and error bars represent 95% confidence intervals about the means (big dots). Significant differences between average diameters of swarms on prey grown at different temperatures are shown; ** $p < 0.01$, *** $p < 0.001$ (Tukey-adjusted contrasts). The dataset for this figure and the R script used to analyze it and make the figure are available on Zenodo (10.5281/zenodo.10214013).

### *M. xanthus* kills *P. fluorescens* reared at 32°C on M9cas agar

Although *M. xanthus* was previously shown to utilize *P. fluorescens* as prey to fuel population growth [18,39] while substantially reducing *P. fluorescens* population size [18], the composition of agar medium used in the earlier experiments differed from the M9cas agar used in the Fig 1 swarming assays. Because both species had been reared at 32°C before being mixed in the earlier experiments, we sought to confirm our expectation that GJV1 kills *P. fluorescens* reared overnight at 32°C on M9cas agar. Indeed, when GJV1 was distributed across the surface of circular *P. fluorescens* lawns reared at 32°C on M9cas, the *P. fluorescens* populations were reduced by approximately 90% relative to control populations lacking *M. xanthus* after 4 days of *M. xanthus* growth (S1 Fig). From this result, we infer that in the experiment reported in Fig 1, *M. xanthus* extensively killed *P. fluorescens* within the areas covered by *M. xanthus* swarms expanding through 32°C-reared *P. fluorescens* lawns.

### *P. fluorescens* reared at lower temperatures kills *M. xanthus*

The observed inhibition of *M. xanthus* growth by lawns of *P. fluorescens* reared at 22°C (Fig 1) might have been caused by either a nonlethal mechanism that merely prevents (or greatly

slows) *M. xanthus* cell division or by killing. To test between these hypotheses, 20-μl aliquots containing approximately $2 \times 10^8$ cells of *M. xanthus* strain DK3470 were spotted onto lawns of *P. fluorescens* previously grown overnight at 12, 22, or 32˚C. *M. xanthus* population size was then estimated approximately 30 minutes and 7 days after *M. xanthus* inoculation. DK3470 was used for this assay rather than GJV1 or the natural isolates because DK3470 disperses more readily due a mutation that prevents production of adhesive extracellular matrix. As part of the same experiment, we also tested whether the initial population size of *P. fluorescens* inoculated to initiate growth prior to meeting *M. xanthus* might impact any effect of *P. fluorescens* on *M. xanthus* population size. To do so, *P. fluorescens* growth was initiated from culture aliquots at 3 inoculum sizes (approximately $10^5$, $10^6$, and $10^7$ cells/flask) the day prior to addition of *M. xanthus*.

Consistent with the swarming results shown in Fig 1, lawns of *P. fluorescens* reared at 32˚C supported the largest *M. xanthus* populations after a week of incubation (Fig 2, Tukey-adjusted contrasts averaged over interaction terms, all $p < 0.01$). Prey inoculum size had no impact on *M. xanthus* population size when *P. fluorescens* had been reared at 32˚C.

In contrast, and also consistent with the Fig 1 swarming results, *P. fluorescens* reared at 22˚C prior to meeting *M. xanthus* did not merely limit *M. xanthus* growth but slaughtered it to extinction, irrespective of initial prey population size (Fig 2 and S2 Table). No viable *M. xanthus* cells were observed at the limit of detection in any of the treatments in which *P. fluorescens* was reared at 22˚C.

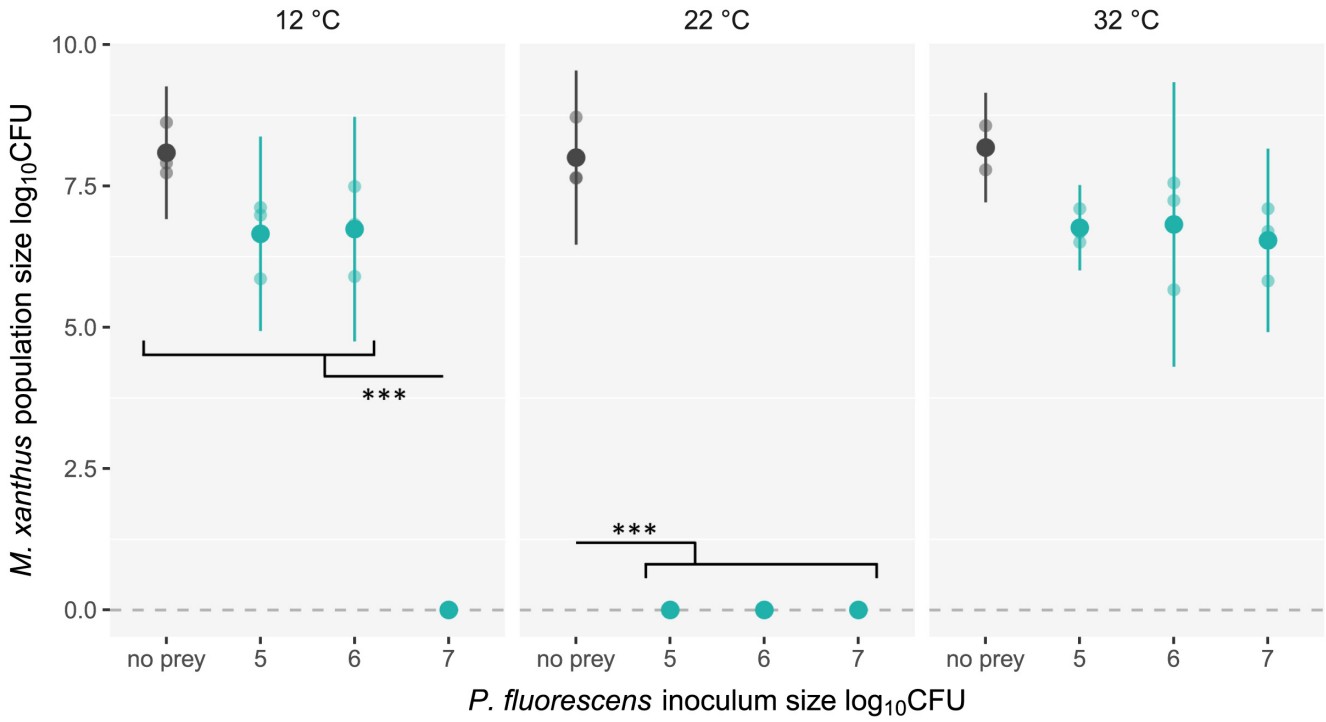

**Fig 2. *P. fluorescens* grown at 22˚C from any inoculum size or at 12˚C from high inoculum size exterminates *M. xanthus*.** *M. xanthus* strain DK3470 population size 7 days after inoculation onto *P. fluorescens* lawns that had grown overnight from one of 3 inoculum population sizes (green dots) at one of 3 temperatures prior to addition of *M. xanthus*. Black dots show corresponding data for controls lacking *P. fluorescens*. Means of log$_{10}$-transformed CFU + 1 values and 95% confidence intervals are shown. Lighter dots are biological replicates ($n = 3$). Significant differences between *M. xanthus* population sizes within prey-growth temperature treatments are shown; *** $p < 0.001$ (Tukey-adjusted contrasts). The dataset for this figure and the R script used to analyze it and make the figure are available on Zenodo (10.5281/zenodo.10214013).

Unlike in the 2 higher-temperature rearing treatments, prey inoculum size greatly impacted *M. xanthus* population size for *P. fluorescens* reared at 12˚C (Fig 2, rearing-temperature × prey-inoculum size interaction term $F_{6,59} = 42.1$, $p < 0.01$; S2 Table). Like *P. fluorescens* reared at 22˚C, some cultures reared at 12˚C also killed *M. xanthus* to extinction but did so only when *P. fluorescens* was inoculated at the highest inoculum size. When *P. fluorescens* was inoculated at the 2 lower inoculum sizes prior to growth at 12˚C, *M. xanthus* grew to population sizes similar to those achieved on *P. fluorescens* reared at 32˚C. Killing of *M. xanthus* by *P. fluorescens* grown at 22˚C (from any inoculum size) or 12˚C (from high inoculum size) occurred rapidly, being observed already after only 30 minutes after inoculation of *M. xanthus* (S2 Fig).

Together, the above results demonstrate both that (i) variation in a single abiotic parameter can determine the direction of killing between a predatory bacterial species and one of its prey species and (ii) such abiotic determination of killing direction can be exerted remotely in time, with the temperature at which *P. fluorescens* grows prior to interaction with *M. xanthus* determining which species kills the other.

## Killing of *M. xanthus* by 12˚C-grown *P. fluorescens* appears contingent on postgrowth population density

We hypothesized that, in the Fig 2 experiment with DK3470, differences in population densities of *P. fluorescens* across treatments might have been at least partially responsible for differences in how *P. fluorescens* grown at different temperatures interacted with DK3470. To examine this hypothesis, we cultivated *P. fluorescens* from the same 3 initial population sizes and at the same 3 temperatures as in the Fig 2 experiment and then assessed population sizes after 22 hours of growth at the 3 temperatures and subsequent incubation at room temperature for 2 hours.

Postgrowth population size of *P. fluorescens* (i.e., *N* after 24 hours total incubation) was independent of initial population size after growth at 32˚C ($R^2 = 0.24$, $F_{1,7} = 2.25$, $p = 0.18$; S3 Fig) but correlated positively with initial population size after growth at 12 and 22˚C ($R^2 = 0.996$, $F_{1,7} = 1709$ for 12˚C; $R^2 = 0.79$, $F_{1,7} = 26.58$ for 22˚C, both $p < 0.01$), with the strongest relationship observed after growth at 12˚C. The sizes of *P. fluorescens* populations grown at 32˚C (from any initial population size) did not differ from those of populations grown at 22˚C from the highest inoculum size. Because all 22˚C-grown populations killed *M. xanthus* but no 32˚C-grown populations did so, our results do not indicate that *P. fluorescens* postgrowth population density has any effect on the efficacy with which 22˚C-grown *P. fluorescens* kills *M. xanthus*, at least over the range observed here.

In contrast, because postgrowth population sizes of 12˚C-grown *P. fluorescens* correlated strongly with initial population sizes and because only those populations initiated at the highest inoculum size killed *M. xanthus* (Fig 2), killing of *M. xanthus* by 12˚C-grown *P. fluorescens* appears to depend positively on postgrowth population density.

## The lethal compound produced by 22˚C-grown *P. fluorescens* is a nonproteinaceous, diffusive secretion

We tested whether the molecules lethal to *M. xanthus* produced by 22˚C-grown *P. fluorescens* are cell-bound or diffusive and whether they are proteinaceous. *P. fluorescens* was grown into lawns overnight at 12, 22, and 32˚C from an initial population size of approximately $10^7$ cells. The resulting lawns were suspended in buffer solution using sterile glass beads and sterile supernatant was extracted after centrifugation and filtration. Supernatant samples were then split; one was heated at 95˚C to test for heat inactivation of protein function, while the other

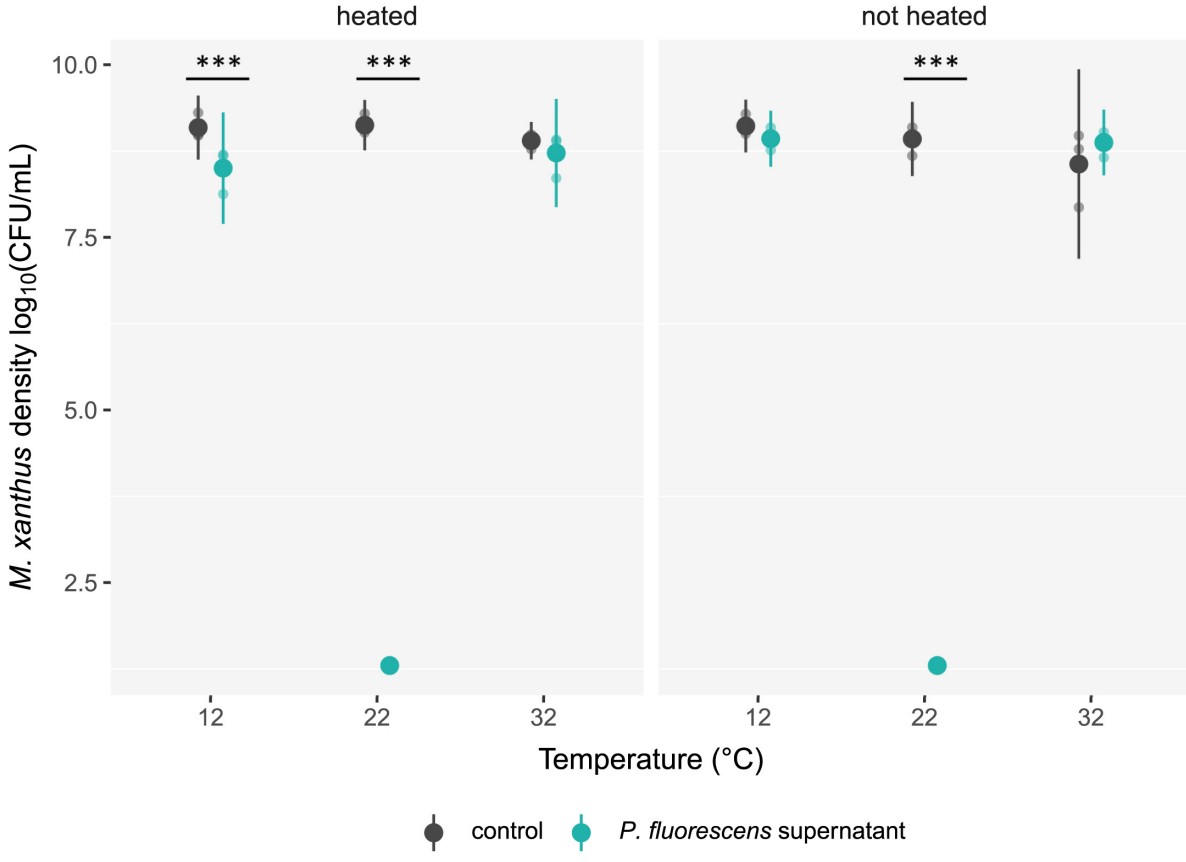

**Fig 3. *P. fluorescens* reared at 22˚C kills *M. xanthus* with a diffusive, nonproteinaceous secretion.** *M. xanthus* density after 6 hours of incubation in the supernatant of buffer suspensions of *P. fluorescens* cultures grown at 12, 22, or 32˚C for 24 hours (green dots) or in control buffer (black dots) and subsequent exposure to either 95˚C or room temperature for 45 minutes (left and right panels, respectively). Means of $\log_{10}$-transformed (CFU + 1)/ml values and 95% confidence intervals are shown. Lighter dots are biological replicates ($n = 3$). Significant differences between predator densities after incubation in supernatant vs. buffer per prey-growth-temperature treatment are shown; *** $p < 0.001$ (Tukey-adjusted contrasts). The dataset for this figure and the R script used to analyze it and make the figure are available on Zenodo (10.5281/zenodo.10214013).

was kept at room temperature. Centrifuged pellets of *M. xanthus* strain DK3470 were then resuspended with the supernatant samples and the resulting suspensions incubated for 6 hours before dilution plating to determine viable population sizes.

Supernatant from 32˚C-grown *P. fluorescens*, whether heated at 95˚C or not, had no effect on DK3470 population size ($F_{11,24} = 441.2$, $p < 0.01$, Tukey-adjusted contrasts $t_{24} = 0.897$, $p = 0.38$ for heated treatment and $t_{24} = −1.567$, $p = 0.13$ for nonheated treatment; Fig 3). Supernatant from 12˚C-grown populations slightly reduced *M. xanthus* population size, at least when heated ($t_{24} = 2.953$, $p < 0.01$ for heated treatment and $t_{24} = 0.92$, $p = 0.37$ for nonheated treatment). Most strikingly, supernatant from 22˚C-grown *P. fluorescens* killed *M. xanthus* populations completely, both when heated and not ($t_{24} = 39.33$ and 38.33 for heated and non-heated treatments, respectively; $p < 0.01$ for both). Because cell-free supernatant from the 22˚C-grown populations effectively killed the predator and 95˚C heat did not impair killing, we infer that the killing agent is a nonproteinaceous, diffusive secretion.

We additionally tested whether the diffusive lethal secretion produced by 22˚C-reared *P. fluorescens* kills *M. xanthus* that had also been reared at 22˚C when the 2 species interact at that same temperature (as well as killing 32˚C-reared *M. xanthus* upon interaction at 32˚C;

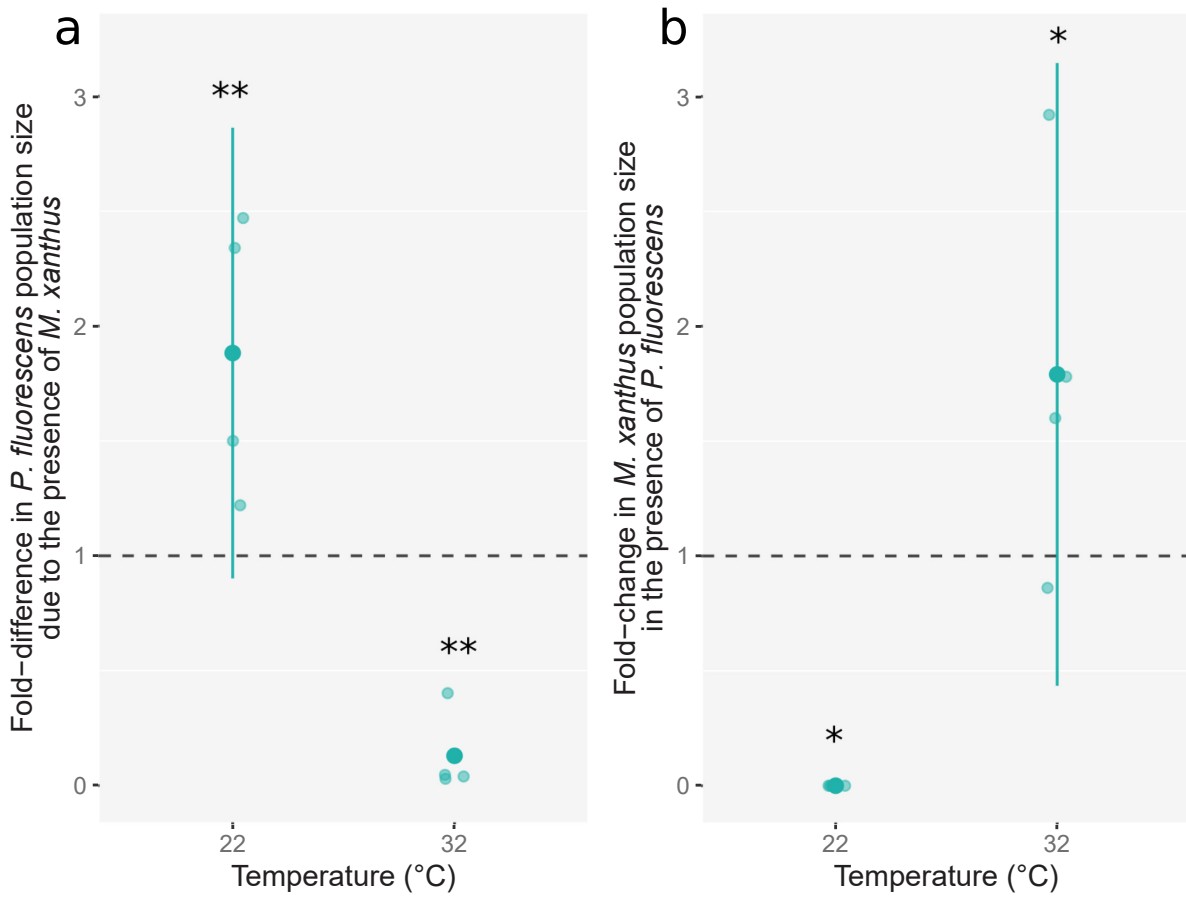

**Fig 4. Predation reversal: *P. fluorescens* grown at 22˚C preys upon *M. xanthus* (a) but *P. fluorescens* grown at 32˚C is preyed upon by *M. xanthus* (b).** Panel (**a**) shows fold-differences in *P. fluorescens* populations to which *M. xanthus* was added relative to control populations to which only liquid buffer was added, when *P. fluorescens* was previously grown at either 22˚C or 32˚C prior interaction. Population sizes were estimated 24 hours after addition of *M. xanthus* or buffer. Panel (**b**) shows fold-change in *M. xanthus* strain DK3470 population size over 24 hours in the presence *P. fluorescens* previously grown at either 22˚C or 32˚C. Lighter dots are biological replicates (*n* = 4). Means of ratios of CFU values and 95% confidence intervals are shown. Asterisks indicate significant differences from 1: * $p < 0.05$ and ** $p < 0.01$ (post hoc two-sided *t* tests against 1 with Benjamini–Hochberg correction for multiple testing). The datasets for this figure and the R script used to analyze them and make the figure are available on Zenodo (10.5281/zenodo.10214013).

S4 Fig). This was indeed the case; no *M. xanthus* cells survived interaction with *P. fluorescens* when rearing of both species and interaction all occurred at 22˚C, whereas *M. xanthus* was unaffected by *P. fluorescens* when rearing of both species and interaction all occurred at 32˚C (S4 Fig; rearing-temperature × *P. fluorescens* presence interaction term $F_{1,12} = 108.93$, $p < 0.001$, Tukey-adjusted contrasts $t_{12} = 14.8$, $p < 0.001$ at 22˚C and $t_{12} = 0.04$, $p = 0.97$ at 32˚C).

## Killed *M. xanthus* fuels *P. fluorescens* population growth

We tested whether 22˚C-reared *P. fluorescens* preys on *M. xanthus*—i.e., consumes nutrients and fuels growth from *M. xanthus* cells it kills—with 2 experiments. Both experiments tested whether *P. fluorescens* can grow on nutrients derived from *M. xanthus* cells killed by 22˚C-reared *P. fluorescens* and one additionally tested whether such nutrients from killed *M. xanthus* are readily diffusible.

In 1 experiment, liquid resuspensions of *M. xanthus* cultures were placed on top of small *P. fluorescens* lawns that had previously grown alone on M9cas agar for 24 hours at either 22 or 32°C. The resulting cocultures, as well as *P. fluorescens*-only controls, were subsequently incubated at 32°C for another 24 hours. Population size estimates of both species were estimated by dilution plating at the start and end of the 24-hour coculture period, allowing us to (i) again test the relative effects of each species on the population size of the other as a function of *P. fluorescens* growth temperature prior to meeting *M. xanthus* and (ii) test whether addition of *M. xanthus* to *P. fluorescens* lawns previously grown at 22°C both results in *M. xanthus* extinction (as already observed) and fuels more *P. fluorescens* population growth than occurs on M9cas alone.

As observed previously, *M. xanthus* both extensively killed and grew in the presence of 32°C-grown *P. fluorescens* but was completely killed by 22°C-grown *P. fluorescens* (Fig 4). Intriguingly, while *M. xanthus* consistently killed >90% of 32°C-reared *P. fluorescens* (Figs 4B and S1), killing of *M. xanthus* by 22°C-grown *P. fluorescens* was even more extensive, completely eliminating the added *M. xanthus* populations at our limit of detection (two-sample two-sided *t* test, $t_3 = 1090.1$, $p < 0.001$). Importantly, 22°C-reared *P. fluorescens* populations were additionally observed to act as predators of *M. xanthus* by growing approximately 80% more in size when *M. xanthus* was added than on M9cas alone (on average $1.53 \times 10^9$ cells [CI 95%: $6.22 \times 108$–$2.44 \times 10^9$] without *M. xanthus* versus $2.65 \times 10^9$ cells [CI 95%: $2.22 \times 10^9$–$3.07 \times 10^9$] with *M. xanthus*, linear model $t = -5.456$, $p < 0.01$; Fig 4A). In other words, *P. fluorescens* cells reared at 22°C underwent significant growth fueled by molecules derived from *M. xanthus* cells that they killed. In so doing, the 22°C-grown *P. fluorescens* cells were predators of *M. xanthus*.

In a separate experiment, cell-free supernatant from suspensions of 22°C-reared *P. fluorescens* lawns was prepared as in the previous experiments and then used to test for *P. fluorescens* growth in 4 treatments. Half of the *P. fluorescens* supernatant was used to resuspend centrifuged pellets of *M. xanthus* strain DK3470; those resuspensions were then incubated for 6 hours to kill *M. xanthus*. Samples from these resuspensions plated on CTT agar after the 6-hour incubation period showed no evidence of *M. xanthus* growth 5 days after plating, confirming that the *P. fluorescens* supernatant killed *M. xanthus* to extinction, as in the previous experiments. No *M. xanthus* cells were added to the other half of the *P. fluorescens* supernatant.

We then inoculated *P. fluorescens* at 2 initial densities (approximately $10^6$ and approximately $10^8$ CFU/ml) into the *P. fluorescens* supernatant to which *M. xanthus* cells had either been added or not and monitored subsequent population sizes. While *P. fluorescens* unsurprisingly grew to some degree on nutrients present in its own supernatant alone (S5 Fig) [52], from both initial densities, it grew to much higher densities in the supernatant containing killed *M. xanthus* (49- and 22-fold higher for the low and high initial *P. fluorescens* density treatments, respectively; $F_{1,8} = 43.11$, $p < 0.01$, Tukey-adjusted contrasts $t_8 = -8.84$, $p < 0.01$; S5 Fig). These results show that nutrients derived from killed *M. xanthus* cells—nutrients sufficient in quantity to fuel extensive *P. fluorescens* growth—had become highly diffusible within 6 hours after *M. xanthus* cells were exposed to supernatant from 22°C-grown *P. fluorescens*.

### *P. fluorescens* supernatant lethal to *M. xanthus* harms a minority of diverse tested species

We began investigating the killing range of the compound(s) lethal to *M. xanthus* produced by 22°C-grown *P. fluorescens* by exposing cells from 7 other bacterial species to the *M. xanthus*-lethal supernatant. We tested both gram-positive and gram-negative species, including cells of

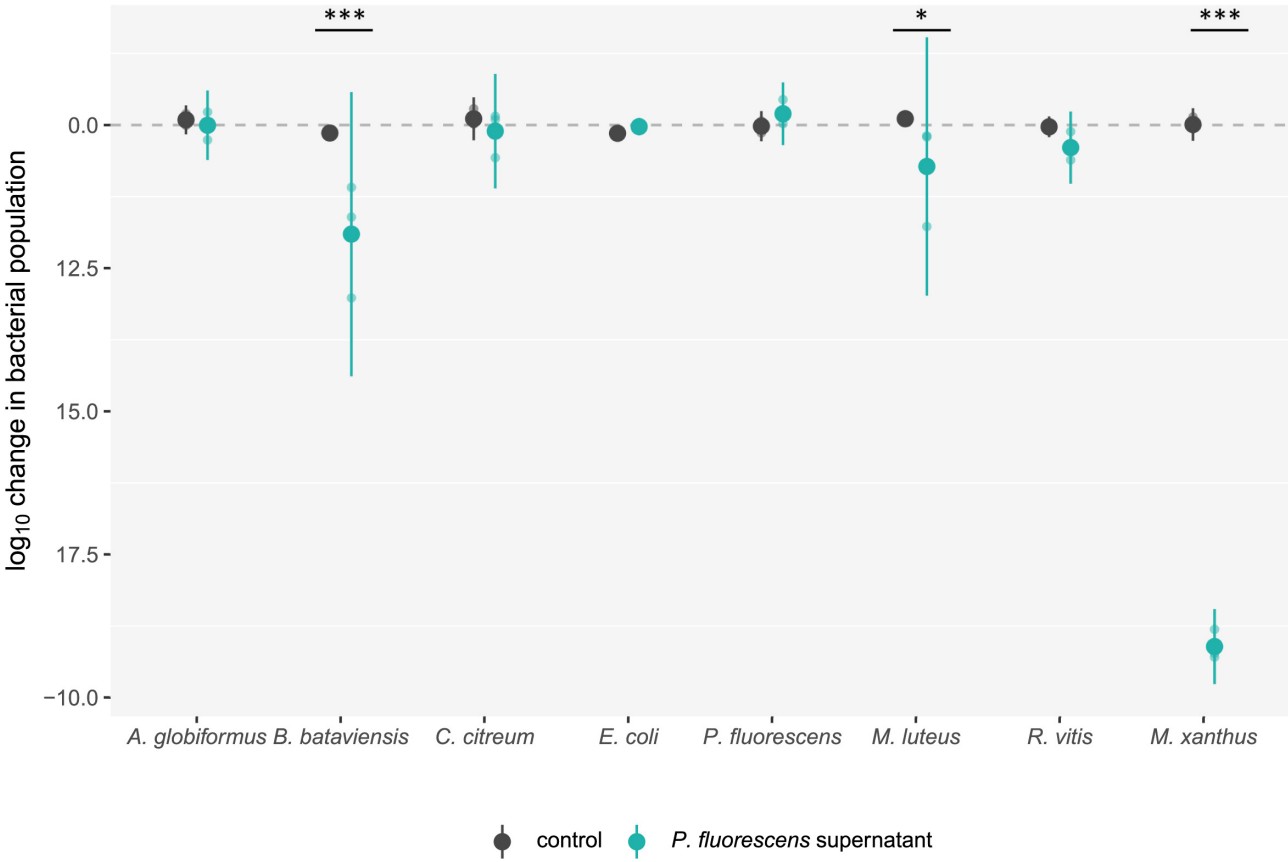

**Fig 5. Secretions of 22˚C-reared *P. fluorescens* harm a minority of diverse tested species.** Change in densities of diverse bacterial species ($\log_{10}$) after 6 hours of incubation in supernatant from a buffer suspension of *P. fluorescens* grown at 22˚C (green dots) or in a control buffer (black dots). Mean values and 95% confidence intervals are shown. Lighter dots are biological replicates ($n = 3$). Significant differences between supernatant and control treatments are shown; *** $p < 0.001$ and * $p < 0.05$ (two-sided *t* tests with Benjamini–Hochberg correction). The dataset for this figure and the R script used to analyze it and make the figure are available on Zenodo (10.5281/zenodo.10214013).

*P. fluorescens* itself as controls. Although *M. xanthus* was again killed effectively in these assays ([Fig 5]; $F_{7,32} = 102.89$, $p < 0.01$, Tukey contrasts on supernatant versus control marginal means $t_{32} = 29.45$ $p < 0.001$), only 2 of the other examined species were significantly harmed by *P. fluorescens* supernatant ([Fig 5]). *B. bataviensis* and *M. luteus* populations exposed to the supernatant were reduced to less than 10% and 50% of their corresponding control populations, respectively ($t_{32} = 5.7$ $p < 0.001$ and $t_{32} = 2.69$ $p < 0.05$, respectively), whereas the other species were not significantly affected (Tukey contrasts on supernatant versus control marginal means, all $p > 0.05$).

## Discussion

The animal world features numerous examples of predation role reversal [53–55] and mutual predation [56–58], and at least some are determined by ecological context [7]. In the microbial world, ecological factors have been shown to greatly impact predation efficiency but, to our knowledge, without reversing the direction of predation. We have presented an extreme example of the importance of the abiotic environment; changing a single physical variable in the growth conditions of one species reverses the character of its subsequent interaction with another. Specifically, changing the temperature at which a bacterial species grows prior to

interspecific interaction flips hierarchy in a food chain with the sometime prey species *P. fluorescens* becoming a predator of the often-predacious species *M. xanthus*. Reversed killing of *M. xanthus* by *P. fluorescens* is mediated by highly effective nonproteinaceous diffusible molecules. However, *P. fluorescens* supernatant containing the killing compound(s) harmed only 3 of 7 examined bacterial species (including *M. xanthus*) and did so in a phylogenetically idiosyncratic manner.

## Mechanism and range of killing

Thoroughly understanding the selective forces shaping production of an antimicrobial compound by bacteria residing in species-rich communities is a complex challenge. This requires quantification of both production costs and of how the compound's antagonistic effects on producers' diverse neighbors collectively impact producer fitness. Such fitness effects include benefits of harming or killing neighbors derived both from securing competed-for resources and from nutrients obtained directly from prey cells. Such thorough understanding would also require characterization of compound features such as energy requirements for its synthesis, how its production is regulated, whether it is cell-bound or diffusible, and, if diffusible, its spatial diffusion ranges across producer habitats and the temporal durability of the compound's antagonistic effects under relevant ecological conditions.

Several of our results are first steps toward understanding the evolutionary character of the compound produced by cool-reared *P. fluorescens* that is lethal to *M. xanthus*. First, because *M. xanthus* survival is highly sensitive to acid stress in a density-dependent manner [25], we asked whether 22˚C-grown *P. fluorescens* might kill *M. xanthus* by acidifying its surroundings. This is not the case, as supernatants from *P. fluorescens* cultures grown at 22˚C and 32˚C did not differ in pH from the buffer control (linear model, $F_{2,6} = 3.83$, $p = 0.08$).

Second, and as previously mentioned, the lethal compound is a diffusible secretion the production of which is regulated by temperature. It is functionally robust after having been heated to 95˚C, indicating that it is not a protein, and a modified drop-collapse assay (as in [59]; S6 Fig) performed with supernatant from our experiments suggests that the lethal compound is a biosurfactant. In our experiments, the compound was produced prior to interaction with *M. xanthus*.

Finally, supernatant from 22˚C-grown *P. fluorescens* not only rapidly killed *M. xanthus* but also rapidly caused sufficient cell decomposition to generate highly diffusible nutrients that could fuel extensive *P. fluorescens* population growth after passing through a 0.2-μm filter (S5 Fig). It is possible that the same compound caused both the death and rapid degradation of *M. xanthus* cells.

The negative effects of *P. fluorescens* supernatant on *B. bataviensis* and *M. luteus* populations (Fig 5) may be caused by the same compound(s) that exterminated *M. xanthus*. If this were the case, our results would indicate that its mechanism of antagonism is unrelated to Gram-type cell-wall specificities; both gram-positive and gram-negative species were harmed by the supernatant and other species of both Gram types were unharmed. Thus, if the same compound(s) kills the 3 sensitive species, the phylogenetic distribution of its killing range appears to be highly idiosyncratic.

## Apparent density-dependent killing

Some known predators such as *M. xanthus* and *Streptomyces* spp. secrete diffusible toxins capable of killing many prey cells at a distance [13,60]. Similarly, temperature-dependent killing of *M. xanthus* by *P. fluorescens* is carried out by a diffusible secretion. Diffusible secretions

are known to mediate density-dependent behaviors in bacteria [61]. It appears that killing of *M. xanthus* by *P. fluorescens* is also density dependent in at least some ecological contexts.

Among *P. fluorescens* populations reared at 12˚C, only populations initiated from the highest starting density in our experiments—which, in turn, reached the highest final density before meeting *M. xanthus*–killed *M. xanthus* (Figs 2 and S3). This suggests that, at low temperatures, a threshold *P. fluorescens* density needs to be reached for secretions lethal to *M. xanthus* to be produced in quantities sufficient to kill extensively. This effect might result simply from a lethal compound increasing linearly with local population size, or compound production might be regulated by a form of quorum sensing. However, high density alone is insufficient to allow predation of *M. xanthus* by *P. fluorescens*; temperature is also determinative. *P. fluorescens* populations initiated at the same density and reaching the same final density were lethal after growth at 22˚C but not at 32˚C (Figs 2 and S3). These results suggest that the efficiency of various forms of microbial killing and predation mediated by diffusible molecules may be positively density dependent, but also suggest that the form or existence of such density dependence may often depend on ecological context.

## Past ecology shapes future interactions

An intriguing feature of our findings is that the observed predator–prey reversal was driven by ecological context prior to, rather than during, predator–prey interaction. In most of our interaction experiments, only the temperature at which one species (*P. fluorescens*) grew prior to meeting the other (*M. xanthus*) varied across treatments; after *P. fluorescens* growth in isolation, agar plates from all 3 growth-temperature treatments were brought to the same temperature before the 2 species were mixed. Yet, differences in *P. fluorescens* biotic environments caused by the differences in growth temperature were not eliminated by the subsequent temperature equalization. *P. fluorescens* reared at 32˚C was extensively killed when exposed to *M. xanthus* (Figs 4 and S1), whereas *P. fluorescens* reared at 22˚C slaughtered *M. xanthus* to extinction (Figs 2, 3, 4, and S2) and used the released nutrients to fuel its own growth (Figs 4 and S5). The temporal separation of the causative ecological factor and the impacted interaction in our experiments has important implications for our understanding of interaction networks in natural microbial communities. Within such communities, microecological features can change rapidly over time and migration across variable microenvironments can be extensive. Thus, the strength and direction of microbial interactions may often be shaped not only by the abiotic context in which they are taking place but also by the recent ecological histories of interacting parties.

## Myxobacterial predator–prey interactomes

When offered as potential prey, diverse bacterial species vary greatly in the degree to which they fuel *M. xanthus* growth [18,39], and some exhibit resistance to predation [62]. The finding that a non-myxobacterial species can be a predator of *M. xanthus* rather than its prey should inform future investigations into myxobacterial predator–prey interactomes. Species exhibiting resistance to predation by myxobacteria should also be examined for the ability to kill and consume them. Abiotic ecology, including variables altered by climate change, can be expected to strongly impact the character of predator–prey interactomes between myxobacterial species and other species in both directions of predation and may sometimes determine the very direction of predation across variable natural habitats.

The ecological dependence of microbial predator–prey interactomes and antagonisms more broadly also has applied significance [63–65]. The range of ecological conditions under which microbial biocontrol agents can be expected to harm their targets should be investigated in planning their use.

## How much microbial warfare results in predation?

*P. fluorescens* has long been known to antagonize other microbes; the species secretes various antibiotics [66] and emits toxic volatile compounds such as cyanide [67] and has been applied as a biocontrol agent to control plant pathogens [68]. Yet, despite its known potential to antagonize and kill other species, including *M. xanthus* [62], it does not appear that *P. fluorescens* has previously been labeled a predator. However, because the strain used here can clearly grow on nutrients derived from cells it has killed (Figs 4 and S4), it is indeed a predator under the broad phenomenological definition of 'predation' focused on organismal interactions that we adopt. In this definition, 'predator' refers to any organism that kills another and then consumes it [69], irrespective of the predator's evolutionary history of predation. The degree to which an organism's ancestral lineage has undergone selection favoring nutrient acquisition from predation is not a criterion. This definition has the advantage of focusing simply on whether a clearly defined interaction occurs, leaving separate the question of that interaction's evolutionary origins, which may often be unclear, especially for microbes.

In light of our results and the ubiquity of intermicrobial killing [29,70,71], predation by microbe-killing microbes that have rarely, if ever, previously been labeled as predators is likely to be more common than is generally considered. This includes killing between conspecifics, which occurs pervasively among *M. xanthus* natural isolates [72–74] and is also common in many other species [29]. Well-studied killing mechanisms, including both contact-dependent secretion systems such as the Type VI system and diverse diffusible toxins that kill remotely [29], may often confer selective benefits by mediating predation as well as by reducing competition. Perhaps most bacterial species engage in predation to some degree. Species frequently labeled as predators likely cluster far along a continuous spectrum of evolutionary adaptedness for predation.

## Supporting information

**S1 Fig. *M. xanthus* kills *P. fluorescens* grown on M9cas agar at 32˚C.** Percentage reduction of *P. fluorescens* population size after 4 days in the presence of *M. xanthus* relative to in the absence of *M. xanthus*. Colors correspond to 3 independent replicates each run with 2 technical replicates. The dataset for this figure and the R script used to analyze it and make the figure are available on Zenodo (10.5281/zenodo.10214013).
(PDF)

**S2 Fig. *P. fluorescens* grown at 22˚C kills *M. xanthus* within 30 minutes of interaction.** *M. xanthus* population sizes 30 minutes after inoculation onto *P. fluorescens* lawns grown overnight from one of 3 inoculum sizes and at one of 3 temperatures (green dots) or onto bacteria-free control plates that had been incubated overnight at one of the same 3 temperatures prey (black dots). Means of $\log_{10}$-transformed CFU + 1 values and 95% confidence intervals are shown. Note that, due to a technical issue with dilution plating for 1 replicate of the 12˚C treatment, the highest available plated dilution was too low to accurately count colonies; the corresponding plates had more colonies than could feasibly be counted. We therefore attributed counts of 1,000 for these plates, which was clearly a substantial underestimate in each case. These underestimated values are identified in the graph with red circles around corresponding data points. Lighter dots are biological replicates (*n* = 3). The dataset for this figure and the R script used to analyze it and make the figure are available on Zenodo (10.5281/zenodo.10214013).
(PDF)

**S3 Fig. Final *P. fluorescens* population size correlates positively with initial population size after growth at 12 and 22˚C but not at 32˚C.** Relationship between *P. fluorescens* initial and

final population sizes after 22 hours of growth at different temperatures 12, 22, or 32˚C. Log$_{10}$-transformed CFU values ($n = 3$), linear fits, and 95% confidence intervals about the linear fits are shown. The dataset for this figure and the R script used to analyze it and make the figure are available on Zenodo (10.5281/zenodo.10214013).
(PDF)

**S4 Fig. *P. fluorescens* kills *M. xanthus* when both species are pregrown and interact at 22˚C but not when both are pregrown and interact at 32˚C.** DK3470 population sizes are shown 24 hours after interaction with *P. fluorescens* when the 2 species interact at the same temperature at which both had been reared prior to interaction. Means of log$_{10}$-transformed CFU + 1 values and 95% confidence intervals are shown. Lighter dots are biological replicates ($n = 3$). \*\*\* $p < 0.001$ (Tukey-adjusted contrasts) for the difference between *M. xanthus* population size after interaction with *P. fluorescens* (green dots) vs. in the control treatment (black dots) when the 2 species were reared and interacted at 22˚C. The dataset for this figure and the R script used to analyze it and make the figure are available on Zenodo (10.5281/zenodo.10214013).
(PDF)

**S5 Fig. Reversed predation decomposes *M. xanthus* cells into diffusible nutrients.** Remains of *M. xanthus* killed by 22˚C-reared *P. fluorescens* sufficient to fuel large *P. fluorescens* population growth pass through 0.2-μm filters within 6 hours of interspecies interaction. Estimated densities (log-transformed CFU/ml, $n = 3$) of *P. fluorescens* populations over time inoculated at 2 initial densities (approximately $10^6$ and approximately $10^8$ CFU/ml) into supernatant from 22˚C-reared *P. fluorescens* lawns to which *M. xanthus* cells were either added (and which killed those *M. xanthus* cells, green dots) or not (black dots). Trend lines show local polynomial regression fitting and dark gray bands represent 95% confidence regions. The dataset for this figure and the R script used to analyze it and make the figure are available on Zenodo (10.5281/zenodo.10214013).
(PDF)

**S6 Fig. Drop-collapse assay with *P. fluorescens* supernatants.** Picture of 20-μl aliquots of supernatants from liquid suspensions of *P. fluorescens* grown on M9cas agar at 32˚C (left) and 22˚C (center) and of M9 buffer control (right). The 3 replicates of this assay yielded visually indistinguishable results; 1 replicate is shown here.
(PDF)

**S1 Table. Statistical analysis of *M. xanthus* swarming data on prey lawns.** Linear model and Type III ANOVA for swarming data using prey identity, predator identity, temperature treatment, and their interactions as explanatory variables. Post hoc contrasts between temperature treatments are computed for each predator–prey combination.
(PDF)

**S2 Table. Statistical analysis of *M. xanthus* growth data.** Linear model and Type III ANOVA for *M. xanthus* growth data using *P. fluorescens* inoculum size, temperature treatment, their interaction, and time as explanatory variables. Post hoc contrasts between inoculum sizes are computed for each temperature–time combination.
(PDF)

## Acknowledgments

We thank August Paula, Jos Kramer, and the Evolutionary Biology group at ETH for helpful discussion.

## Author Contributions

**Conceptualization:** Marie Vasse, Francesca Fiegna, Ben Kriesel, Gregory J. Velicer.

**Formal analysis:** Marie Vasse, Gregory J. Velicer.

**Investigation:** Marie Vasse, Francesca Fiegna, Ben Kriesel, Gregory J. Velicer.

**Methodology:** Marie Vasse, Francesca Fiegna, Ben Kriesel, Gregory J. Velicer.

**Resources:** Gregory J. Velicer.

**Supervision:** Marie Vasse, Francesca Fiegna, Gregory J. Velicer.

**Visualization:** Marie Vasse.

**Writing – original draft:** Marie Vasse, Francesca Fiegna, Ben Kriesel, Gregory J. Velicer.

**Writing – review & editing:** Marie Vasse, Francesca Fiegna, Ben Kriesel, Gregory J. Velicer.

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
