## [Editor Report · Decision Letter 0]

17 May 2023

Dear Dr. Vasse, 

Thank you for submitting your manuscript entitled "Killer prey: Temperature reverses future bacterial predation" for consideration as a Research Article by PLOS Biology.

Your manuscript has now been evaluated by the PLOS Biology editorial staff, as well as by an academic editor with relevant expertise, and I am writing to let you know that we would like to send your submission out for external peer review as a Short Report.

Short Reports present the results from a limited set of experiments that can generally be summarized in 3-4 figures or fewer. The outcomes should be self contained, rather than fitting within the narrative arc of a larger research project or article.

Once your full submission is complete, your paper will undergo a series of checks in preparation for peer review. After your manuscript has passed the checks it will be sent out for review. To provide the metadata for your submission, please Login to Editorial Manager (https://www.editorialmanager.com/pbiology) within two working days, i.e. by May 19 2023 11:59PM.

Kind regards,

Paula

---

Senior Editor

PLOS Biology

---

## [Decision Letter · Decision Letter 1]

4 Jul 2023

Dear Dr. Vasse,

Thank you for your patience while your manuscript "Killer prey: Temperature reverses future bacterial predation" was peer-reviewed at PLOS Biology. It has now been evaluated by the PLOS Biology editors, an Academic Editor with relevant expertise, and by several independent reviewers. 

In light of the reviews, which you will find at the end of this email, we would like to invite you to revise the work to thoroughly address the reviewers' reports.

As you will see below, the reviewers find the work interesting, but they all have concerns that will need to be solved before further consideration. We think that it is important that you include novel experiments in which the predation assays with predator & prey grown at the same temperatures (ie, pre-incubation of both species at the same temperature as the predation assays are performed at), and include an experiment in which killing by & growth of P. fluorescens is done in the same experiment, i.e. currently, you show killing by P. fluorescens in one experiment and then in a different experiment show that P. fluorescens can grow on the supernatant from the lysed Myxococcus cells. We also think that either including the data for the drop collapse experiment mentioned in the Discussion or not mention this experiment at all, and including more information from the literature about the antimicrobial activity of P. fluorescent would be important. Please also address the other reviewers' points by editing the text/title. 

Given the extent of revision needed, we cannot make a decision about publication until we have seen the revised manuscript and your response to the reviewers' comments. Your revised manuscript is likely to be sent for further evaluation by all or a subset of the reviewers.

**IMPORTANT - SUBMITTING YOUR REVISION**

*Re-submission Checklist*

*Published Peer Review*

*PLOS Data Policy*

*Blot and Gel Data Policy*

Sincerely,

Paula

---

Senior Editor

PLOS Biology

REVIEWS:

Reviewer #1: Myxococcus.

Reviewer #2: Bacterial competition.

Reviewer #3: Bacterial predation behaviour.

Reviewer #1: The manuscript by Vasse et al. describes a study of predation between Pseudomonas fluorescens and Myxococcus xanthus, which changes sign depending on what temperature the P. fluorescens was reared. The authors show that cold-grown P. fluorescens produces a diffusible non-proteinaceous compound which kills M. xanthus, and P. fluorescens is able to grow on the nutrients released by killed M. xanthus.

The manuscript is well-written, the experiments are described clearly and the conclusions are supported by the data. 

Did the authors do predation assays with M. xanthus and P. fluorescens at the three different temperatures? ie with pre-incubation of both species at the same temperature the predation assays were then performed at? It seemed strange to start with a setup where pre-incubation temperatures were varied and the predation assays temperatures weren't. 

Similarly, does P. fluorescens still prey upon M. xanthus at 22 oC rather than the other way round, if both organisms were pre-incubated at 32 oC? 

How long does the pre-incubation acclimation last?

Specific Comments

I found the title a bit difficult to get my head around, and only really understood what the authors meant after reading the abstract. It might be worth considering a longer but clearer title.

Line 120. Spell out rH as relative humidity

Line 149. It wasn't clear until quite a lot later why flasks were being used rather than petri dishes. Might make sense to explain the rationale earlier?

Line 226. To harvest P. fluorescens supernatant, ... 

Lines 251-4. To what extent does the pre-growth of prey lawns reduce the concentration of free nutrients for M. xanthus growth? These few lines seem a bit vague.

Line 258 'addition' might be better than 'release'.

Reviewer #2: Authors present an interesting study with the main finding of temperature-dependent predation evasion of Myxococcus xanthus by Pseudomonas fluorescens. The authors also make a claim of inversion of predator-prey relationship at the temperature that P. fluorescens kills M. xanthus, however the evidence presented is not strong enough for how emphatic these claims are. So while the overall study is interesting and should be published, the claims of predation inversion need to be reduced considerably. The discussion is especially problematic and must be remedied prior to publication.

Otherwise, the paper is reasonably clearly written and the figures support all other conclusions well. The discussion hinges strongly on the predation reversal claim, of which the evidence presented is not strong enough to warrant this level of certainty. In the discussion it is also claimed that the killing molecules (no evidence whether it is one or multiple compounds mediating the killing) is "extremely effective", however no serial dilution experiment was done, just the 30 minute exposure experiment with no earlier time points. This speed of killing is quite fast, but is not used to follow up on the possible nature of the killing substance. Additionally, in the discussion a result about a drop-collapse experiment is mentioned, but these data are not included in the results section. In general, the discussion section is too expansive and speculative. It should be shortened substantially and edited to focus on the results presented. For example, the section about 'past ecology shaping future interactions': no co-culture experiments were presented which could have tested whether temperature also determined the nature of the interaction when the two species are grown together, instead the results are extensively re-described and speculated upon. Additionally, many statements in the discussion are presented without appropriate references. For example: L460 "Within such communities, micro-ecological features can change rapidly over time and migration across variable microenvironments can be extensive."

Specific Comments

Predator genotype, prey identity and temperature affect swarm diameter, and interaction. While the effect of P. fluorescens is dramatic, the other effects are minor and should be presented as such.

The reduction of swarming radius when prey is present vs not, implies that a carbon free medium that forces the predator to consume prey to swarm at all may have been a better control, as the current implication is that prey presences reduces swarming.

This sentence is very confusing and should be re written to be more understandable.

"We hypothesized that, in the Fig. 2 experiment with DK3470, population densities of P. fluorescens after overnight growth might have mediated some effects of P. fluorescens on DK3470 growth or death, whether partially or fully."

Did you check the pH of the P. fluorescens supernatants from different temperatures? If the pH is the same then this would support the killing of M. xanthus being caused by a more interesting molecule. The speed of killing implies this could be pH mediated.

Just because P. fluorescens can grow on the supernatant from killed M. xanthus, doesn't mean that the killing by P. fluorescens is predatory. This evidence is not strong enough to say that the direction of predation is temperature dependent. If it could be shown that P. fluorescens kills and eats M. xanthus in the absence of any other nutrient source at 22C, then this would be definitive evidence that the direction of predation is changed. However this experiment needs to be done with both species co-cultured, and to truly be definitive it would be nice to see radionuclide labeled M. xanthus incorporated into P. fluorescens, though this is extremely high level of evidence. Co-culturing without exogenous nutrients and P. fluorescens growing better would be enough to support predation.

The observation that P. fluorescens supernatant is only lethal to M. xanthus is very interesting. It would also be interesting to know if it can kill eukaryotic predators like Tetrahymena or Acanthamoeba.

There is a section of the discussion about a drop-collapse experiment to show that the killing substance is a surfactant, but these data are not in the results. Either this section needs to be removed or the data presented.

This sentence (Line 454-458) is misleading. "Yet differences in P. fluorescens biotic environments caused by the differences in growth temperature were not eliminated by the subsequent temperature equalization. P. fluorescens reared at 32 °C was extensively killed when exposed to M. xanthus (Fig. S1), whereas P. fluorescens reared at 22 °C slaughtered M. xanthus to extinction (Figs. 2, 3 and S2) and used the released nutrients to fuel its own growth (Fig. 4)." This implies that the evidence for P. fluorescens "fueling its own growth" on killed M. xanthus comes from the same experiment as the killing of M. xanthus, which is not the case. This needs to be fixed before publication as it greatly overstates the evidence presented.

Other problematic statements: L468 "The finding that some bacteria can be predators of myxobacteria". The evidence presented isn't sufficient to claim a single predator of M. xanthus, nevermind some predators of myxobacteria in general.

This section adds very little of interest: "The ecological dependence of microbial predator-prey interactomes and antagonisms more broadly also has applied significance (63-65). The range of ecological conditions under which microbial biocontrol agents can be expected to harm their targets should be investigated in planning their use."

"How much microbial warfare results in predation?". This section is generally problematic. The evidence presented in this paper does not show that P. fluorescens is preying upon the M. xanthus, since the way the experiment was done was highly artificial. It doesn't even fit the authors definition since the P. fluorescens that killed the M. xanthus are not the same cells that consumed the nutrients, so technically this would be scavenging.. The next paragraph then also remains highly speculative: far more evidence is required to show that other bacteria that may incidentally consume nutrients derived from killed competitors are predators. Myxobacteria and BALOs are considered specialized bacterial predators due to large bodies of evidence, such evidence is not even close to existing for other bacteria.

Reviewer #3: Comments to the authors

In their manscript, Vasse et al. describe the influence of an abiotic factor, i.e. temperature, to antagonistic interbacterial interactions. Specifically, they focus on the predator-prey relationship of the predatory bacterium Myxococcus xanthus with Pseudomonas fluorescence. From several quantitative co-culture experiments the authors deduce that P. fluorescence reared at 32C serves as prey for M. xanthus, while P. fluorescens reared at lower temperature (22C) acts as a predator on M. xanthus. Further experiments with P. fluorescens culture supernatant suggests that not-proteinaceous diffusible molecules are effective in killing M. xanthus.

The experiments are sound, data are well presented and evaluated, which includes appropriate statistics. The significance of the observations are discussed in the broader context of the ecological relevance of interspecies interactions and how they are shaped by environmental changes.

The findings are very interesting and will promote new thoughts on our understanding of bacterial antagonistic interactions, specifically predation. Especially the observation that P. fluorescence grows on nutrients from M. xanthus alone challenges our definition of bacterial predation, which might be a much more common strategy than currently recognized.

My main comment is that the study should relate more to known literature. Antimicrobial activity of Pseudomonas fluorescence has been demonstrated before, and the species as been described as a biocontrol agent promoting plant growth by inhibiting (fungal) plant pathogens (see for example the review Haas, D., Défago, G. Nat Rev Microbiol 3, 307-319 (2005).; also, there are lots of studies on the interaction with e.g. Ralstonia sp.). 

It is also known that P. fluorescence produces siderophores and antibiotics, but also volatile toxic compounds, such as cyanide, which should be discussed in the context of the presented observations. Even more, it was shown that cyanide production depends on growth phase of P. fluorescence (e.g. Askeland RA, Morrison SM Appl Environ Microbiol. 1983)

So, the question arises whether the observed temperature-dependent effect of P. fluorescence may really show a dependence on growth phase, as temperature obviously affects growth rate. This is could be relevant not only in the context of cyanide, but also of other antimicrobial agents. 

I believe the authors could come up with an experiment to test this - I consider it important, as it might alter the way they discuss their observations.

Specific comments:

- p. 7, l. 4: "LB 0.5% agar". Does 0.5% refer to NaCl concentration, or agar concentration (soft agar)?

- p. 17, l. 422-423: The authors mention a "drop-collapse assay" suggesting the antimicrobial compound is a biosufactant. Is the experiment included in the manuscript or not? If these are preliminary observations it should please be stated as such, and "data not shown"

- p. 11, l. 261: "across the dataset, swarm sizes were impacted by [.....] the rearing temperature." It looks to me that only for P. fluorescence has an impact?

---

## [Decision Letter · Decision Letter 2]

8 Nov 2023

Dear Dr. Vasse,

Thank you for your patience while we considered your revised manuscript "Killer prey: Pre-interaction ecology reverses bacterial predation" for consideration as a Short Reports at PLOS Biology. Your revised study has now been evaluated by the PLOS Biology editors, the Academic Editor, and two of the original reviewers. 

In light of the reviews, which you will find at the end of this email, we are pleased to offer you the opportunity to address the remaining points from the reviewers in a revision that we anticipate should not take you very long. We will then assess your revised manuscript and your response to the reviewers' comments with our Academic Editor aiming to avoid further rounds of peer-review, although might need to consult with the reviewers, depending on the nature of the revisions.

In particular, we think that the additional experiments mentioned by reviewer #2 are not necessary for publication. We think it is important that you address the points raised by reviewer #3 and that you clarify/edit the text to address the criticisms raised by reviewer #2.

Please also address the following editorial and policy requests:

1. DATA POLICY:

A) Supplementary files (e.g., excel). Please ensure that all data files are uploaded as 'Supporting Information' and are invariably referred to (in the manuscript, figure legends, and the Description field when uploading your files) using the following format verbatim: S1 Data, S2 Data, etc. Multiple panels of a single or even several figures can be included as multiple sheets in one excel file that is saved using exactly the following convention: S1_Data.xlsx (using an underscore).

B) Deposition in a publicly available repository. Please also provide the accession code or a reviewer link so that we may view your data before publication. 

Regardless of the method selected, please ensure that you provide the individual numerical values that underlie the summary data displayed in the following figure panels as they are essential for readers to assess your analysis and to reproduce it: Figures 1, 2, 3, 4, 5, and Supplementary Figures S1, S2, S3, S4, S5, S6.

2. CODE POLICY

Per journal policy, as the code that you have generated is important to support the conclusions of your manuscript, we require that you make it available without restrictions upon publication. Please ensure that the code is sufficiently well documented and reusable, and that your Data Statement in the Editorial Manager submission system accurately describes where your code can be found.

Please note that sole deposition of data or code to GitHub would not be compliant with our policies, as this could be changed after publication (https://journals.plos.org/plosbiology/s/data-availability). However, once the data/code is final, you can archive your publicly available GitHub data to Zenodo. Once you do this, it will also generate a DOI number that you can provide us with. See the process for doing this here: https://docs.github.com/en/repositories/archiving-a-github-repository/referencing-and-citingcontent

3. Please provide a blurb which (if accepted) will be included in our weekly and monthly Electronic Table of Contents, sent out to readers of PLOS Biology, and may be used to promote your article in social media. The blurb should be about 30-40 words long and is subject to editorial changes. It should, without exaggeration, entice people to read your manuscript. It should not be redundant with the title and should not contain acronyms or abbreviations.

4. Please clarify your financial statement "The author(s) received no specific funding for this work." Please note that we require financial information that has allowed you to do this research even if it was not specifically granted to do this work. 

5. We suggest a change in the title: "The temperature at which bacteria grow before interacting can reverse the direction of predation between species ".

**IMPORTANT - SUBMITTING YOUR REVISION**

*Resubmission Checklist*

*Published Peer Review*

*PLOS Data Policy*

*Blot and Gel Data Policy*

Sincerely,

Paula

---

Senior Editor

PLOS Biology

REVIEWS:

Reviewer #2: This version is somewhat improved but I still think the evidence for predation inversion is insufficient. The authors clearly think the evidence is much stronger, so either this should be published to let readers decide or the editor can decide how strong the evidence presented is.

I also think that for a short report the discussion is far too long and strays away from discussing the results of the presented work into a perspective on antagonism and predation, which is a detraction from the work.

Why I think the evidence is insufficient:

1. Experiments showing that M. xanthus kills P. fluorescens at 32C (Fig S1) are clear, but accompanying evidence that predation is also occurring is insufficient. Figure 1 shows mostly a reduction in swarm diameters on lawns of other bacteria. References 18 and 40 appear to use agar without 0.3% casitone, so the evidence for predation is much better. It's unclear why 0.3% casitone plates were used in the present study. The response to my previous comment indicates that M. xanthus is indeed using the casitone for part of its growth which makes it harder to conclude the amount of predation it is engaging in.

In Figure 4 the increase of M xanthus suffers from the same lack of controls as described below.

2. Experiments showing that P. fluorescens kills M. xanthus at 22C are clear, but the evidence for predation is insufficient. As far as I can tell in the methods for the experiment presented in Figure 4, there is no control to show that the modest increase in P. fluorescens (~2-fold) after 24h after addition of M. xanthus is due to predation. Two controls are missing: one where nothing is added to the P. fluorescens lawn, and one where the same amount of M9 buffer the M. xanthus cells were resuspended in is added. Based on the text in the results maybe this was done? But in the methods it is not described.

3. It is clear that the 22C supernatant from P. fluorescens lyses M. xanthus cells, providing nutrients for growth. 

4. The discussion of the density-dependence of the killing factor production has issues:

The logic that because the 22C grown cultures all had comparable density to the 32C cultures, but killing of M. xanthus only occurs at 22C, therefore density doesn't affect killing is undermined by the observation that density does affect killing at 12C. It seems likely that lower initial inocula at 22C would result in less growth and at some point, affect killing. So as currently written, the results do not support the conclusion. Either lower inocula densities at 22C need to be tested, or it should be concluded that killing of M. xanthus is density-dependent. 

This is stated in the discussion, so the results section should be edited to make this conclusion clearer.

The paragraph around Line 530 is predicated on the lack of experiments done at 22C with lower densities which would allow the conclusion that the killing molecule is density dependent, but not produced at 32C.

Further notes on the evidence for predation by P. fluorescens

In the methods section "Tests for growth of P. fluorescens on nutrients from M. xanthus killed by P. fluorescens and for effects of P. fluorescens rearing temperature on M. xanthus DK3470 and P. fluorescens population sizes after interaction"

There does not appear to include a control where just 20uL M9 was added on top of the P. fluorescens lawns, or nothing. So, it's hard to conclude that the growth that occurred in the next 24h of the lawns where M. xanthus was added is indeed due to nutrients released from killing.

But in the results:

Line 438 "to act as predators of M. xanthus by growing ~80% more in size when M. xanthus was added than on M9cas alone". 

This does not match the methods description around line 240, nor the caption of Figure 4.

If M9 with cas was added this isn't a good control since the cas could be used for growth. The experiment should be M. xanthus cells in M9 salts, or just M9 salts, or nothing. Then compare the growth after 24h. If P. fluorescens grows more with M xanthus cells then it killed and ate them and predation is well supported.

Further notes about the discussion being too long

The discussion is very long and proscriptive, especially for a short report.

The paragraph around line 540 could be greatly shortened.

The paragraph around line 560 feels very strong given the weakness of the predation inversion evidence as described above.

The paragraph around line 570 veers into unrelated territory, as the presented study does not address antagonism, since the message is clearly about predation. Overall, this paragraph reads as a perspective, pushing the notion that interference competition frequently involves predation. This does not belong in a short report.

Other issues in this paragraph:

"P. fluorescens has not often, if ever, been described as a predator". This should be answerable in the literature. I expect the answer is never.

"because the strain used here can clearly grow on nutrients derived from cells it has killed". Again, this is not clear from the evidence presented. Figure 4 lacks controls.

Other remarks

73 "Predatory weapons" this is confusing terminology. Weapons should be used to describe systems used for interference competition between peers, not predatory adaptations. As in ref 29.

Line 303. Sequester is not the right word to use here. The lawn bacteria are using the casitone for anabolism, not just taking it up and holding on to it.

Line 525

Instead of the review reference 61, the exact references for M. xanthus and Streptomyces secreting diffusible killing molecules and eating prey should be provided. 

The supernatant should also be treated by proteinase K to affirm its non-proteinaceous nature. This is a pretty standard assay.

Figure 4 Caption mentions lighter dots, but I only see the one dot with lines. Either the caption needs editing or the plot corrected.

I appreciate the inclusion of the drop-collapse data, but looking at Figure S6 I can't see any difference between the three plates. Either an arrow needs to be added or a picture from a different angle or with a black background or something because this is not clear at all.

Reviewer #3: I find that the authors have further improved their very interesting manuscript: they amended the text and added new experiments, or argued convincingly when they decided against addressing a specific point. I have only two minor points to be addressed, outlined below.

New Figure 4 and corresponding text: This new experiment nicely supports the main claims. However, the figure only shows the fold-change in population size but gives no information about the absolute population sizes, nor (in the methods section) on the number of M. xanthus cells added to P. fluorescence. To better interpret to what extent M. xanthus fuels growth of P. fluorescence, I would ask the authors to please include this information.

Figure S6: As requested, the authors have added an image showing the drop-collapse assay mentioned in the discussion. However, in the provided image and legend, I can't make out what the figure is supposed to show. From the mentioned reference, I understand that it should be liquid drops on a lid of a 96-well plate (?) I would please ask for a better quality image and a bit more information in the figure legend.

---

## [Editor Report · Decision Letter 3]

27 Nov 2023

Dear Dr. Vasse,

Thank you for your patience while we considered your revised manuscript "Killer prey: Ecology reverses bacterial predation" for publication as a Short Reports at PLOS Biology. This revised version of your manuscript has been evaluated by the PLOS Biology editors and the Academic Editor.

Based on our Academic Editor's assessment of your revision, we are likely to accept this manuscript for publication, provided you satisfactorily address the following data and other policy-related requests.

1. DATA POLICY:

A) Supplementary files (e.g., excel). Please ensure that all data files are uploaded as 'Supporting Information' and are invariably referred to (in the manuscript, figure legends, and the Description field when uploading your files) using the following format verbatim: S1 Data, S2 Data, etc. Multiple panels of a single or even several figures can be included as multiple sheets in one excel file that is saved using exactly the following convention: S1_Data.xlsx (using an underscore).

B) Deposition in a publicly available repository. Please also provide the accession code or a reviewer link so that we may view your data before publication.

Regardless of the method selected, please ensure that you provide the individual numerical values that underlie the summary data displayed in the following figure panels as they are essential for readers to assess your analysis and to reproduce it: Figures 1, 2, 3, 4, 5, and Supplementary Figures S1, S2, S3, S4, S5, S6.

2. CODE POLICY

Per journal policy, as the code that you have generated is important to support the conclusions of your manuscript, we require that you make it available without restrictions upon publication. Please ensure that the code is sufficiently well documented and reusable, and that your Data Statement in the Editorial Manager submission system accurately describes where your code can be found.

Please note that sole deposition of data or code to GitHub would not be compliant with our policies, as this could be changed after publication (https://journals.plos.org/plosbiology/s/data-availability). However, once the data/code is final, you can archive your publicly available GitHub data to Zenodo. Once you do this, it will also generate a DOI number that you can provide us with. See the process for doing this here: https://docs.github.com/en/repositories/archiving-a-github-repository/referencing-and-citingcontent

3. We suggest a change in the title: "The temperature at which bacteria grow before interacting can reverse the direction of predation between species ".

We expect to receive your revised manuscript within two weeks. 

*Published Peer Review History*

*Press*

Sincerely,

Paula

---

Senior Editor,

pjaureguionieva@plos.org,

PLOS Biology

---

## [Editor Report · Decision Letter 4]

30 Nov 2023

Dear Dr Vasse,

Thank you for the submission of your revised Short Reports "Killer prey: Ecology reverses bacterial predation" for publication in PLOS Biology. On behalf of my colleagues and the Academic Editor, Lotte Søgaard-Andersen, I am pleased to say that we can in principle accept your manuscript for publication, provided you address any remaining formatting and reporting issues. These will be detailed in an email you should receive within 2-3 business days from our colleagues in the journal operations team; no action is required from you until then. Please note that we will not be able to formally accept your manuscript and schedule it for publication until you have completed any requested changes.

PRESS

Sincerely, 

Paula

---

Senior Editor

PLOS Biology
